# Different coexistence patterns between apex carnivores and mesocarnivores based on temporal, spatial, and dietary niche partitioning analysis in Qilian Mountain National Park, China

Wei Cong[1], Jia Li[2], Charlotte Hacker[3], Ye Li[1], Yu Zhang[1], Lixiao Jin[1], Yi Zhang[1], Diqiang Li[1], Yadong Xue[1]*, Yuguang Zhang[1]*

[1]Ecology and Nature Conservation Institute, Chinese Academy of Forestry, Key Laboratory of Biodiversity Conservation, State Forestry and Grassland Administration, Beijing, China; [2]Institute of Ecological Conservation and Restoration, Chinese Academy of Forestry, Beijing, China; [3]Department of Biological Sciences, Duquesne University, Pittsburgh, United States

**\*For correspondence:**
xueyadong334@163.com (YX);
yugzhang@sina.com.cn (YZ)

**Competing interest:** The authors declare that no competing interests exist.

**Abstract** Carnivores play key roles in maintaining ecosystem structure and function as well as ecological processes. Understanding how sympatric species coexist in natural ecosystems is a central research topic in community ecology and biodiversity conservation. In this study, we explored intra- and interspecific niche partitioning along spatial, temporal, and dietary niche partitioning between apex carnivores (wolf *Canis lupus*, snow leopard *Panthera uncia*, Eurasian lynx *Lynx lynx*) and mesocarnivores (Pallas's cat *Otocolobus manul*, red fox *Vulpes vulpes*, Tibetan fox *Vulpes ferrilata*) in Qilian Mountain National Park, China, using camera trapping data and DNA metabarcoding sequencing data. Our study showed that apex carnivore species had more overlap temporally (coefficients of interspecific overlap ranging from 0.661 to 0.900) or trophically (Pianka's index ranging from 0.458 to 0.892), mesocarnivore species had high dietary overlap with each other (Pianka's index ranging from 0.945 to 0.997), and apex carnivore and mesocarnivore species had high temporal overlap (coefficients of interspecific overlap ranging from 0.497 to 0.855). Large dietary overlap was observed between wolf and snow leopard (Pianka's index = 0.892) and Pallas's cat and Tibetan fox (Pianka's index = 0.997), suggesting the potential for increased resource competition for these species pairs. We concluded that spatial niche partitioning is likely to key driver in facilitating the coexistence of apex carnivore species, while spatial and temporal niche partitioning likely facilitate the coexistence of mesocarnivore species, and spatial and dietary niche partitioning facilitate the coexistence between apex and mesocarnivore species. Our findings consider partitioning across temporal, spatial, and dietary dimensions while examining diverse coexistence patterns of carnivore species in Qilian Mountain National Park, China. These findings will contribute substantially to current understanding of carnivore guilds and effective conservation management in fragile alpine ecosystems.

## eLife assessment

This paper provides an **important** assessment of competition dynamics allowing coexistence of the carnivore guild within a large national park in China. Multiple surveying techniques (camera traps and DNA metabarcoding) provide **convincing** evidence that spatial segregation represents the main

strategy of coexistence, while species have a certain degree of temporal and dietary overlap. Altogether, the manuscript provides information critical to the conservation and management agenda of the park.

## Introduction

Carnivores play key roles in maintaining ecosystem structure and function, as well as ecological processes (*Ripple et al., 2014*). Apex carnivores are often classically specialized hunters occupying top trophic positions that work to suppress the number of herbivores and mesocarnivores through predation, competition, and trophic cascades (*Ripple et al., 2014*; *Newsome et al., 2017*; *Ritchie and Johnson, 2009*). Understanding how sympatric carnivore species coexist in natural ecosystems is a central focus in community ecology and biodiversity conservation (*Chu et al., 2017*). Human activities and climate change are causing large-scale habitat loss and fragmentation, posing significant challenges to carnivore interspecies relationships (*Li et al., 2021*; *Manlick and Pauli, 2020*; *Ripple et al., 2014*). Such relationships are becoming increasingly intricate, leading to greater research efforts aimed at determining the strategies that facilitate coexistence despite intense competition (*Kuijper et al., 2016*; *Smith et al., 2017*).

The competitive exclusion principle dictates that species with similar ecological requirements are unable to successfully coexist (*Hardin, 1960*; *Gause, 1934*). Thus, carnivores within a guild occupy different ecological niches based on a combination of three niche dimensions, i.e., spatial, temporal, and trophic (*Schoener, 1974*). Spatially, carnivore species within the same geographical area exhibit distinct distributions that minimize overlap in resource use and competition. For example, carnivores can partition habitats based on habitat feature preferences and availability of prey (*de Satgé et al., 2017*; *Garrote and Pérez de Ayala, 2019*; *Goldyn et al., 2003*; *Strampelli et al., 2023*). Temporally, differences in seasonal or daily activity patterns among sympatric carnivores can reduce competitive interactions and facilitate coexistence. For example, carnivores can exhibit temporal segregation in their foraging behaviors, such as diurnal versus nocturnal activity, to avoid direct competition (*Finnegan et al., 2021*; *Nasanbat et al., 2021*; *Searle et al., 2021*). Trophically, carnivore species can diversify their diets to exploit different prey species or sizes, thereby reducing competition for food resources. For example, carnivores can exhibit dietary specialization to optimize their foraging efficiency and minimize competitive pressures (*Steinmetz et al., 2021*). Currently, research on the niche partitioning of sympatric carnivores primarily focuses on one or two dimensions, with little attention paid to all three (*Alexander et al., 2016*; *Karanth et al., 2017*; *Li et al., 2022a*; *Santos et al., 2019*; *Shao et al., 2021*; *Strampelli et al., 2023*; *Tsunoda et al., 2020*; *Vilella et al., 2020*). This limitation restricts the multi-scale understanding of coexistence mechanisms among carnivores.

In recent years, camera trapping and DNA metabarcoding technology has been widely used in wildlife monitoring and research. Camera trapping enables the monitoring of elusive species in remote areas (*Boitani, 2016*; *Palencia et al., 2022*), and has become a helpful tool for accumulating large amounts of time-recorded data that can provide detailed information surrounding animal behavior and activity patterns, which is widely used to investigate species-interspecies relationships (*Frey et al., 2017*). For example, *Li et al., 2019*, used camera trap data to conclude that temporal segregation is a key mechanism for promoting the coexistence of tigers (*Panthera tigris*) and leopards (*Panthera pardus*). DNA metabarcoding provides a noninvasive molecular tool that is more accurate than traditional dietary analysis methods, such as microhistology (*Deiner et al., 2017*; *Newmaster et al., 2013*). DNA metabarcoding technology has been applied to many studies and has offered profound insight into the ecology, conservation, and biological monitoring of rare and endangered species (*Deagle et al., 2019*; *Kartzinel et al., 2015*). For example, *Shao et al., 2021*, found that dietary niche partitioning promoted the coexistence of species in the mountains of southwestern China based on DNA metabarcoding. The combination of camera trapping and DNA metabarcoding can result in robust data for exploring the mechanisms of coexistence surrounding carnivore guilds.

The Qilian Mountains constitute a biodiversity hotspot with one of the richest carnivore assemblages in the world (*Di Minin et al., 2016*). To understand the competition and coexistence among different carnivore species in this area, we explored the habitat use, activity patterns, and prey item composition of sympatric carnivore species comprised of apex carnivores and mesocarnivores across Qilian Mountain National Park using camera trap data and DNA metabarcoding data. Based on

theories surrounding resource partitioning and niche differentiation, as well as studies on interactions and coexistence among carnivorous species (*Haswell et al., 2018*; *Linnell and Strand, 2000*), we hypothesized that differentiation along one or more niche axes is beneficial for the coexistence of the carnivorous guild in the Qilian Mountains. We expected that spatial niche differentiation promotes the coexistence of large carnivores in the Qilian Mountain region, as they are more likely than small carnivores to spatially avoid interspecific competition (*Davis et al., 2018*). Mesocarnivores may coexist either spatially or temporally due to increased interspecific competition for similar prey (*Di Bitetti et al., 2010*; *Donadio and Buskirk, 2006*). Dietary niche differentiation may be a significant factor for promoting coexistence between large and mesocarnivore species due to differences in body size (*Gómez-Ortiz et al., 2015*; *Lanszki et al., 2019*).

## Results

### Sympatric carnivore identification

Of the 480 scat samples sequenced, those which had no sequencing data were inconclusive, consisted of non-target species, or host species with low sample sizes (one Asian badger [*Meles leucurus*] and four upland buzzard [*Buteo hemilasius*]) were removed (*Figure 3—figure supplement 1*). The remaining 404 scat samples were composed of three apex carnivores (49 wolf, 147 snow leopard, 19 Eurasian lynx) and three mesocarnivores (63 Pallas's cat, 87 red fox, 39 Tibetan fox).

### Spatial distribution difference and overlap

A total of 322 camera trap sites were surveyed after relocating infrared cameras that did not capture any target carnivore species. A total of three cameras were considered to have failed due to loss. We analyzed data from 319 camera sites and obtained 14,316 independent detections during a total effort of 37,192 effective camera trap days. We recorded wolf in 26 sites, snow leopard in 109 sites, Eurasian lynx in 36 sites, red fox in 92 sites, and Tibetan fox in 34 sites. However, the camera detection rates of Pallas's cat were too low to analyze species occupancy and daily activity patterns.

According to the findings derived from single-season, single-species occupancy models, the snow leopard demonstrated a notably higher probability of occupancy overall compared to other carnivore species, estimated at 0.437 (*Table 1*). Conversely, the Eurasian lynx exhibited a lower occupancy probability, estimated at 0.161. Further analysis revealed that the occupancy probabilities of the wolf and Eurasian lynx declined with increasing normalized difference vegetation index (ndvi) (*Table 2*, *Figure 1*). Additionally, wolf occupancy probability displayed a negative relationship with roughness index and a positive relationship with prey availability. Snow leopard occupancy probabilities exhibited a negative relationship with distance to roads and ndvi. In contrast, both red fox and Tibetan fox demonstrated a positive relationship with distance to roads. Moreover, red fox occupancy probability increased with higher human disturbance and greater prey availability.

The detection probabilities of wolf, snow leopard, red fox, and Tibetan fox exhibited an increase with elevation (*Table 2*). Moreover, there was a positive relationship between the detection probability of Tibetan fox and prey availability. The detection probabilities of snow leopard and Eurasian lynx declined as human disturbance increased.

The Sørensen similarity index (*S*) ranged from 0.1 to 0.5 (*Table 3*). Compared with other combinations of apex-mesocarnivore pairs, snow leopard and red fox (*S*=0.477) had relatively high spatial overlap, while Eurasian lynx and Tibetan fox (*S*=0.198) had the lowest spatial overlap. Moreover, spatial overlap of apex versus apex carnivores and mesocarnivores versus mesocarnivores was relatively low.

### Daily activity patterns and differences

A total of 1444 independent records were obtained for five carnivore species, consisting of 79 records of wolf, 458 records of snow leopard, 126 records of Eurasian lynx, 421 records of red fox, and 359 records of Tibetan fox. Among apex carnivores (*Table 3*, *Figure 2*), the daily activity was similar between snow leopard and Eurasian lynx and their diel activity overlap was close to 1 ($\Delta_4$=0.900, p=0.285), their daily activity peak was at 21:00 hr and dawn. However, the wolf had a significantly different daily activity pattern with snow leopard ($\Delta_4$=0.676, p<0.001) and Eurasian lynx ($\Delta_4$=0.661, p<0.001), and its daily activity peak happened around 9:00 and 18:00 hr. Tibetan fox and red fox had

different activity patterns and peaks (p<0.001). The activity peak for red fox activity peaked at 3:00 and 21:00 hr, while the Tibetan fox had a prolonged active bout between noon and dusk. Temporal activity patterns between apex carnivores and mesocarnivores were significantly different, except for wolf and Tibetan fox ($\Delta_4$=855, p=0.118).

## Dietary composition, diversity, and similarity

A total of 26 unique prey species were identified from 9 taxonomic orders (*Figure 3*, *Supplementary file 1a*). Artiodactyla and lagomorpha were the most frequently detected in the diets of apex carnivores and mesocarnivores, accounting for 32.81% and 70.18% of prey counts, respectively (*Figure 3—figure supplement 2*). Blue sheep made up 26.50% of prey counts in apex carnivore diet, while plateau pika made up 67.11% of prey counts in mesocarnivore diet. Livestock were present in 17.98% of apex carnivore diet counts and were present in 4.82% mesocarnivore diet counts.

The dietary niche overlap among all carnivore species can be found in *Table 3*. Wolf and snow leopard had the highest dietary niche overlap value among apex carnivores ($O_{jk}$ = 0.892). The value of Pianka's index was generally low between apex carnivores and mesocarnivores, except wolf and red fox ($O_{jk}$ = 0.811). In contrast, observed dietary overlap was greatest among the mesocarnivores, especially Pallas's cat and Tibetan fox, with a value of 0.997.

Red fox had the greatest richness of prey with a value of 16, while Pallas's cat and Tibetan fox had the lowest diversity of prey with richness value of 6 (*Supplementary file 1b*). Dietary similarity was assessed using inversed Jaccard's distances. Diets were most similar between wolf and snow leopard with a value of 0.588 and least similar between Pallas's cat and Tibetan fox with a value of 0.200. All other pairs fell between the values of 0.2 and 0.5 (*Supplementary file 1c*).

## Discussion

Our study addresses, for the first time, the coexistence patterns of carnivore species present in Qilian Mountain National Park across multiple dimensions of niche partitioning. This work substantially contributes to current understanding of carnivore guilds and offers helpful information for biodiversity conservation at a regional scale. Moreover, our study provides important insights into the potential mechanisms of niche partitioning among sympatric carnivores, particularly intra- and interspecific relations between apex carnivores and mesocarnivores. Specifically, we found that the overall trend of spatial overlap across carnivores is relatively low, that apex carnivores overlap in time and diet, that mesocarnivores showed a high degree of dietary overlap, and that there was substantial similarity in diel activity patterns between apex carnivores and mesocarnivores. These results indicate that carnivores with similar ecological traits foster co-occurrence by adjusting their daily activity patterns and using differing food resources to minimize competitive interactions.

We found dietary and temporal overlap among apex carnivores, showing that spatial partitioning is responsible for their successful coexistence in this area, consistent with our hypothesis. Wolf and snow leopard had the highest dietary overlap and prey similarity between apex carnivore pairs in this study, showing that the avoidance of space and time plays an important role in their coexistence. Recent evidence proves that habitat preference facilitates the coexistence of wolf and snow leopard (*Shrotriya et al., 2022*). Their hunting strategies may be impacted by their habitat selection. Solitary snow leopards are more suitable for hiding in habitats with features that favor ambush predators, while wolves hunt in packs (*Shrotriya et al., 2022*). It is clear that wild ungulates (e.g. blue sheep) constituted the primary proportion of wolf and snow leopard diet, followed by small mammals such as plateau pika, Himalayan marmot (*Marmota himalayana*), and woolly hare (*Lepus oiostolus*). In addition, livestock consumption also contributed to the high degree of overlap in their diets (*Wang et al., 2014*). This supports the optimal foraging theory, in which large predators preferentially select food resources that provide maximum benefit (*Brown et al., 1999*), but also showed that greater competition for resources is likely to occur between wolf and snow leopard due to their use of the same prey species in this area. This may be especially true in times of habitat stress when resources are poor.

Snow leopard and Eurasian lynx had the highest temporal overlap between apex carnivore pairs in our case, showing that spatial and dietary partitioning facilitate their coexistence. Eurasian lynx is considered an opportunistic predator, and its prey varies among different regions with its primary dietary resource being ungulates and small mammals. For example, Eurasian lynx showed a strong

**Table 1.** Summary of occupancy rate and detection probability of different species for the optimal models (ΔAIC≤2).
Data from *Table 1—source data 1*.

| Species | Models | Number of parameters | AIC | ΔAIC | AIC Wt | ψ | p |
|---------|--------|----------------------|-----|------|--------|---|---|
| | *Psi* (rix+ndvi+prey); *P* (ele+prey) | 7 | 282.14 | 0.000 | 0.077 | 0.180 | 0.153 |
| | *Psi* (rix+ndvi+prey); *P* (ele+hdis) | 7 | 282.45 | 0.302 | 0.066 | 0.183 | 0.145 |
| | *Psi* (rix+ndvi+prey); *P* (ele) | 6 | 283.67 | 1.525 | 0.036 | 0.217 | 0.111 |
| | *Psi* (rix+ndvi+prey); *P* (ele+hdis+prey) | 8 | 284.10 | 1.959 | 0.029 | 0.179 | 0.154 |
| | *Psi* (ele+rix+ndvi+prey); *P* (ele+prey) | 8 | 284.12 | 1.972 | 0.029 | 0.168 | 0.165 |
| | *Psi* (rix+ndvi+disrd+prey); *P* (ele+prey) | 8 | 284.14 | 2.000 | 0.028 | 0.178 | 0.156 |
| Wolf | **Model average** | | | | | **0.184** | **0.147** |
| | *Psi* (disrd+ndvi); *P* (hdis+prey) | 6 | 970.69 | 0.000 | 0.035 | 0.436 | 0.419 |
| | *Psi* (ele+disrd); *P* (hdis+prey) | 6 | 970.81 | 0.113 | 0.033 | 0.435 | 0.421 |
| | *Psi* (.); *P* (hdis+prey) | 4 | 970.99 | 0.292 | 0.030 | 0.437 | 0.420 |
| | *Psi* (disrd+ndvi); *P* (ele+hdis+prey) | 7 | 971.12 | 0.429 | 0.028 | 0.440 | 0.413 |
| | *Psi* (ele+disrd+ndvi); *P* (hdis+prey) | 7 | 971.13 | 0.439 | 0.028 | 0.433 | 0.420 |
| | *Psi* (.); *P* (ele+hdis+prey) | 5 | 971.21 | 0.511 | 0.027 | 0.442 | 0.413 |
| | *Psi* (disrd); *P* (ele+hdis+prey) | 6 | 971.22 | 0.525 | 0.027 | 0.444 | 0.410 |
| | *Psi* (disrd); *P* (hdis+prey) | 5 | 971.52 | 0.825 | 0.023 | 0.437 | 0.420 |
| | *Psi* (ele); *P* (hdis+prey) | 5 | 971.62 | 0.928 | 0.022 | 0.435 | 0.421 |
| | *Psi* (ndvi); *P* (hdis+prey) | 5 | 971.64 | 0.947 | 0.022 | 0.437 | 0.420 |
| | *Psi* (ele+disrd); *P* (ele+hdis+prey) | 7 | 971.92 | 1.222 | 0.019 | 0.436 | 0.416 |
| | *Psi* (ndvi); *P* (ele+hdis+prey) | 6 | 972.42 | 1.726 | 0.015 | 0.440 | 0.414 |
| | *Psi* (ele+disrd+ndvi); *P* (ele+hdis+prey) | 8 | 972.45 | 1.757 | 0.014 | 0.435 | 0.416 |
| | *Psi* (disrd+ndvi+rix); *P* (hdis+prey) | 7 | 972.67 | 1.974 | 0.013 | 0.436 | 0.419 |
| Snow leopard | **Model average** | | | | | **0.437** | **0.417** |

*Table 1 continued on next page*

*Table 1 continued*

| Species | Models | Number of parameters | AIC | ΔAIC | AIC Wt | ψ | p |
|---|---|---|---|---|---|---|---|
| | *Psi* (ndvi); *P* (hdis) | 4 | 371.32 | 0.000 | 0.025 | 0.163 | 0.321 |
| | *Psi* (rix+ndvi); *P* (hdis) | 5 | 371.40 | 0.076 | 0.024 | 0.161 | 0.322 |
| | *Psi* (ndvi); *P* (hdis+prey) | 5 | 371.64 | 0.325 | 0.021 | 0.170 | 0.296 |
| | *Psi* (rix+ndvi); *P* (hdis+prey) | 6 | 371.86 | 0.539 | 0.019 | 0.167 | 0.300 |
| | *Psi* (ndvi+hdis); *P* (prey) | 5 | 371.94 | 0.623 | 0.019 | 0.163 | 0.315 |
| | *Psi* (ndvi+disrd); *P* (hdis) | 5 | 372.02 | 0.702 | 0.018 | 0.163 | 0.321 |
| | *Psi* (rix+ndvi+hdis); *P* (prey) | 6 | 372.23 | 0.907 | 0.016 | 0.161 | 0.318 |
| | *Psi* (ndvi+hdis); *P* (.) | 4 | 372.24 | 0.916 | 0.016 | 0.156 | 0.343 |
| | *Psi* (rix+ndvi+hdis); *P* (.) | 5 | 372.36 | 1.044 | 0.015 | 0.154 | 0.345 |
| | *Psi* (rix+ndvi+disrd); *P* (hdis) | 6 | 372.39 | 1.072 | 0.015 | 0.161 | 0.321 |
| | *Psi* (ndvi+disrd); *P* (hdis+prey) | 6 | 372.64 | 1.318 | 0.013 | 0.168 | 0.300 |
| | *Psi* (ndvi+prey); *P* (hdis) | 5 | 372.67 | 1.352 | 0.013 | 0.163 | 0.320 |
| | *Psi* (rix+ndvi+prey); *P* (hdis) | 6 | 372.99 | 1.672 | 0.011 | 0.160 | 0.323 |
| | *Psi* (ndvi+disrd+hdis); *P* (.) | 5 | 373.06 | 1.744 | 0.011 | 0.156 | 0.344 |
| | *Psi* (ndvi+hdis); *P* (hdis) | 5 | 373.09 | 1.775 | 0.010 | 0.156 | 0.325 |
| | *Psi* (ndvi+disrd+hdis); *P* (prey) | 6 | 373.11 | 1.792 | 0.010 | 0.162 | 0.319 |
| | *Psi* (rix+ndvi+disrd); *P* (hdis+prey) | 7 | 373.12 | 1.796 | 0.010 | 0.165 | 0.303 |
| | *Psi* (rix+ndvi+hdis); *P* (hdis) | 6 | 373.23 | 1.913 | 0.010 | 0.154 | 0.326 |
| | *Psi* (ele+ndvi); *P* (hdis) | 5 | 373.32 | 1.999 | 0.009 | 0.164 | 0.319 |
| Eurasian lynx | **Model average** | | | | | **0.161** | **0.320** |
| | *Psi* (disrd+hdis+prey); *P* (.) | 5 | 894.44 | 0.000 | 0.039 | 0.369 | 0.391 |
| | *Psi* (disrd+hdis+prey); *P* (ele) | 6 | 894.59 | 0.152 | 0.037 | 0.370 | 0.387 |
| | *Psi* (disrd+hdis+ndvi+prey); *P* (.) | 6 | 895.30 | 0.864 | 0.026 | 0.370 | 0.391 |
| | *Psi* (disrd+hdis+ndvi+prey); *P* (ele) | 7 | 895.37 | 0.934 | 0.025 | 0.370 | 0.387 |
| | *Psi* (disrd+hdis+prey+rix); *P* (.) | 6 | 895.99 | 1.544 | 0.018 | 0.369 | 0.392 |
| | *Psi* (disrd+hdis+prey+rix); *P* (ele) | 7 | 896.16 | 1.717 | 0.017 | 0.370 | 0.388 |
| | *Psi* (disrd+hdis+prey); *P* (prey) | 6 | 896.29 | 1.850 | 0.016 | 0.368 | 0.388 |
| | *Psi* (disrd+hdis+prey); *P* (hdis) | 6 | 896.37 | 1.932 | 0.015 | 0.368 | 0.394 |
| Red fox | **Model average** | | | | | **0.369** | **0.390** |
| | *Psi* (ele+disrd+prey+rix); *P* (ele+prey) | 8 | 326.43 | 0.000 | 0.066 | 0.147 | 0.236 |
| | *Psi* (disrd+prey+rix); *P* (ele+prey+hdis) | 8 | 327.14 | 0.714 | 0.046 | 0.277 | 0.135 |
| | *Psi* (ele+disrd+hdis+prey+rix); *P* (ele+prey) | 9 | 327.20 | 0.775 | 0.045 | 0.151 | 0.230 |
| | *Psi* (ele+disrd+rix); *P* (ele+prey) | 7 | 327.29 | 0.866 | 0.043 | 0.158 | 0.219 |
| | *Psi* (ele+disrd+prey+rix); *P* (ele+prey+hdis) | 9 | 327.53 | 1.100 | 0.038 | 0.153 | 0.222 |
| | *Psi* (ele+disrd+prey); *P* (ele+prey) | 7 | 327.69 | 1.263 | 0.035 | 0.149 | 0.234 |
| | *Psi* (disrd+hdis+prey+rix); *P* (ele+prey) | 8 | 328.01 | 1.585 | 0.030 | 0.260 | 0.142 |
| | *Psi* (ele+disrd); *P* (ele+prey) | 6 | 328.42 | 1.986 | 0.024 | 0.159 | 0.219 |
| Tibetan fox | **Model average** | | | | | **0.182** | **0.205** |

disrd – distance to roads, ele – elevation, ndvi – normalized difference vegetation index, rix – roughness index, hdis – human disturbance.

The online version of this article includes the following source data for table 1:

**Source data 1.** Data captured by camera traps on carnivore species.

preference for brown hare *Lepus europaeus* in Turkey, edible dormice *Glis glis* in Slovenia and Croatia, and chamois *Rupicapra rupicapra* or roe deer *Capreolus capreolus* in Switzerland (***Mengüllüoğlu et al., 2018***; ***Krofel et al., 2011***; ***Molinari-Jobin et al., 2007***). Varied prey selection may be related to sex, age, population density, and season (***Mengüllüoğlu et al., 2018***; ***Odden et al., 2006***). Our results show that woolly hare make up the majority of the Eurasian lynx diet, followed by blue sheep. Woolly hare is mainly distributed in shrubland, meadow, desert, and wetland, while blue sheep tend to choose highly sheltered areas, close to bare rocks and cliffs as habitat. Therefore, prey preferences among snow leopard and Eurasian lynx also contribute to spatial avoidance.

Mesocarnivores had substantial overlap in diet, underscoring food resources as a primary competitive factor that necessitates spatial and temporal partitioning for successful coexistence. This finding aligns with previous research indicating that mesocarnivores use temporal and spatial segregation to reduce competition and the probability of antagonistic interspecific encounters (***Ferreiro-Arias et al., 2021***; ***Li et al., 2022a***). Differences in habitat preference may lead to spatial niche partitioning among mesocarnivores (***Wang et al., 2022***). In addition, species can adjust temporal periods of behavior to respond to environmental change, competition, and predation risk (***Gallo et al., 2022***; ***Finnegan et al., 2021***; ***van der Vinne et al., 2019***). Pallas's cat hunts during crepuscular and diurnal periods and inhabits meadow habitat with greater prey abundance (***Anile et al., 2021***; ***Greco et al., 2022***; ***Ross et al., 2019***). In contrast, red fox is primarily nocturnal and occupies diverse habitats depending on prey abundance (***Goldyn et al., 2003***; ***Pandolfi et al., 1997***; ***Reshamwala et al., 2022***). Tibetan fox is a diurnal hunter of the Tibetan plateau, preferring shrub meadow, meadow steppe, and alpine meadow steppe (***Gong and Hu, 2003***). It is worth noting the substantial overlap in diet between Pallas's cat and Tibetan fox. The dietary overlap between the two was extremely high, with a Pianka's value close to 1. Dietary analyses showed that pika contributed to more than 85% of their collective diets, with 90% of Pallas's cat diet being pika. Pika may be an optimal prey item in the area because of size and year-round activity (***Ross, 2009***). Previous studies have shown that the Pallas's cat and Tibetan fox are specialist predators of pikas (***Harris et al., 2014***; ***Ross, 2009***). However, specialization on pika is facultative in that Pallas's cat and Tibetan fox can select other prey items when pika availability is low (***Harris et al., 2014***; ***Ross, 2009***). This was observed in our study, even though dietary diversity was low.

Apex carnivores and mesocarnivores exhibited considerable overlap in time overall, showing that spatial and dietary partitioning may play a large role in facilitating their coexistence. As confirmed by previous research, kit foxes (*Vulpes macrotis*) successfully coexisted with dominant carnivores by a combination of spatial avoidance and use of alternative resources (***Lonsinger et al., 2017***). Differences in body mass may play a crucial role in minimizing dietary overlap, effectively reducing interspecific competition between apex and mesocarnivores. Of exception in our study, however, was wolf and red fox, who exhibited more dietary overlap, showing that temporal and spatial avoidance may promote their coexistence. As canid generalist-opportunist species, the wolf and red fox consumed similar prey, albeit the red fox may have obtained livestock and ungulate species via scavenging or by preying on very young individuals (***Hacker et al., 2022***). In addition, the wolf and red fox had different peak activity times, suggesting temporal segregation as a potentially strong driver of coexistence. Our occupancy model also revealed that mesocarnivores prefer habitats that coincide with human disturbance, possibly contributing to spatial niche differentiation with other apex carnivore species like the wolf. Our findings echo other recent research which supports the idea that red foxes can coexist with wolves by exploiting a broader ecological niche (***Shrotriya et al., 2022***).

Our study highlights the effectiveness of combining camera trapping with DNA metabarcoding for detecting and identifying both cryptic and rare species within a sympatric carnivore guild. This integrated approach allowed us to capture a more comprehensive view of species presence and interactions compared to traditional visual surveys, whereas it is important to acknowledge the challenges associated with this technique, including the high costs of equipment and the need for specialized training and computational resources to manage and analyze the large volumes of sequence data. Despite these challenges, the benefits of this combined method in improving biodiversity assessments and understanding species coexistence outweigh the drawbacks. However, several restrictions remain for this research. The first limitation involves differences in samples sizes. Although the scat samples of Tibetan fox were relatively low, the accuracy of DNA metabarcoding in informing species presence in diet ensures that data are informative and thus still important for species conservation

**Table 2.** Covariates influencing carnivore occupancy rate and detection probability based on the optimal models (ΔAIC≤2).

| Species | Model component | Covariates | Estimate (β) | SE | Z | p |
|---|---|---|---|---|---|---|
| Wolf | Occupancy | Intercept | –2.031 | 0.497 | 4.083 | <0.001*** |
| | | ndvi | –0.843 | 0.287 | 2.935 | 0.003** |
| | | prey | 1.318 | 0.534 | 2.470 | 0.014* |
| | | rix | –1.166 | 0.570 | 2.046 | 0.041* |
| | Detection | Intercept | –1.920 | 0.457 | 4.198 | <0.001*** |
| | | ele | 0.618 | 0.266 | 2.320 | 0.020* |
| Snow leopard | Occupancy | Intercept | –0.256 | 0.143 | 1.783 | 0.075 |
| | | disrd | –0.163 | 0.172 | 0.953 | 0.341 |
| | | ndvi | –0.095 | 0.145 | 0.653 | 0.514 |
| | Detection | Intercept | –0.451 | 0.124 | 3.628 | <0.001*** |
| | | hdis | –1.830 | 0.520 | 3.519 | <0.001*** |
| | | ele | 0.661 | 0.247 | 2.678 | 0.007** |
| Eurasian lynx | Occupancy | Intercept | –1.879 | 0.316 | 5.941 | <0.001*** |
| | | ndvi | –0.583 | 0.201 | 2.899 | 0.004** |
| | Detection | Intercept | –0.999 | 0.375 | 2.665 | 0.008** |
| | | hdis | –1.785 | 1.690 | 1.056 | 0.291 |
| Red fox | Occupancy | Intercept | –0.441 | 0.177 | 2.488 | 0.013* |
| | | disrd | 0.331 | 0.178 | 1.866 | 0.062 |
| | | hdis | 1.333 | 0.611 | 2.184 | 0.029* |
| | | prey | 0.665 | 0.319 | 2.083 | 0.037* |
| | Detection | Intercept | –0.451 | 0.121 | 3.729 | <0.001*** |
| | | ele | 0.057 | 0.093 | 0.606 | 0.545 |
| Tibetan fox | Occupancy | Intercept | –2.087 | 0.643 | 3.247 | <0.001*** |
| | | disrd | 0.955 | 0.321 | 2.973 | 0.003** |
| | | ele | 0.768 | 0.547 | 1.403 | 0.161 |
| | Detection | Intercept | –1.679 | 0.685 | 2.453 | 0.014* |
| | | ele | 0.882 | 0.417 | 2.112 | 0.035* |
| | | prey | 0.369 | 0.167 | 2.216 | 0.027* |

The different superscript letters represent significance, ***p<0.001, **0.001<p<0.01, *0.01<p<0.05.

management decisions (*Hacker et al., 2022*). Second, the methodology of foraging (e.g. predation or scavenging) and the condition of the prey item (e.g. age or size) cannot be identified in dietary studies (*Hacker et al., 2022*). Pika is a prime component of diet among mesocarnivores, especially in the diet of the Pallas's cat and the Tibetan fox. We surmise that the simultaneous dependence on pika led to partial overlap in spatial and activity patterns, resulting in increased potential competitive interactions. Due to the lack of spatial and temporal analysis of Pallas's cat in our study, further monitoring is needed to develop a comprehensive conservation plan. Despite these limitations, our study provides a foundation from which future studies interested in niche partitioning among carnivores along spatial, temporal, and dietary dimensions can be modeled.

In summary, our study has shown that the coexistence of carnivore species in the landscapes of Qilian Mountain National Park can be facilitated along three niche axes, with spatial segregation being

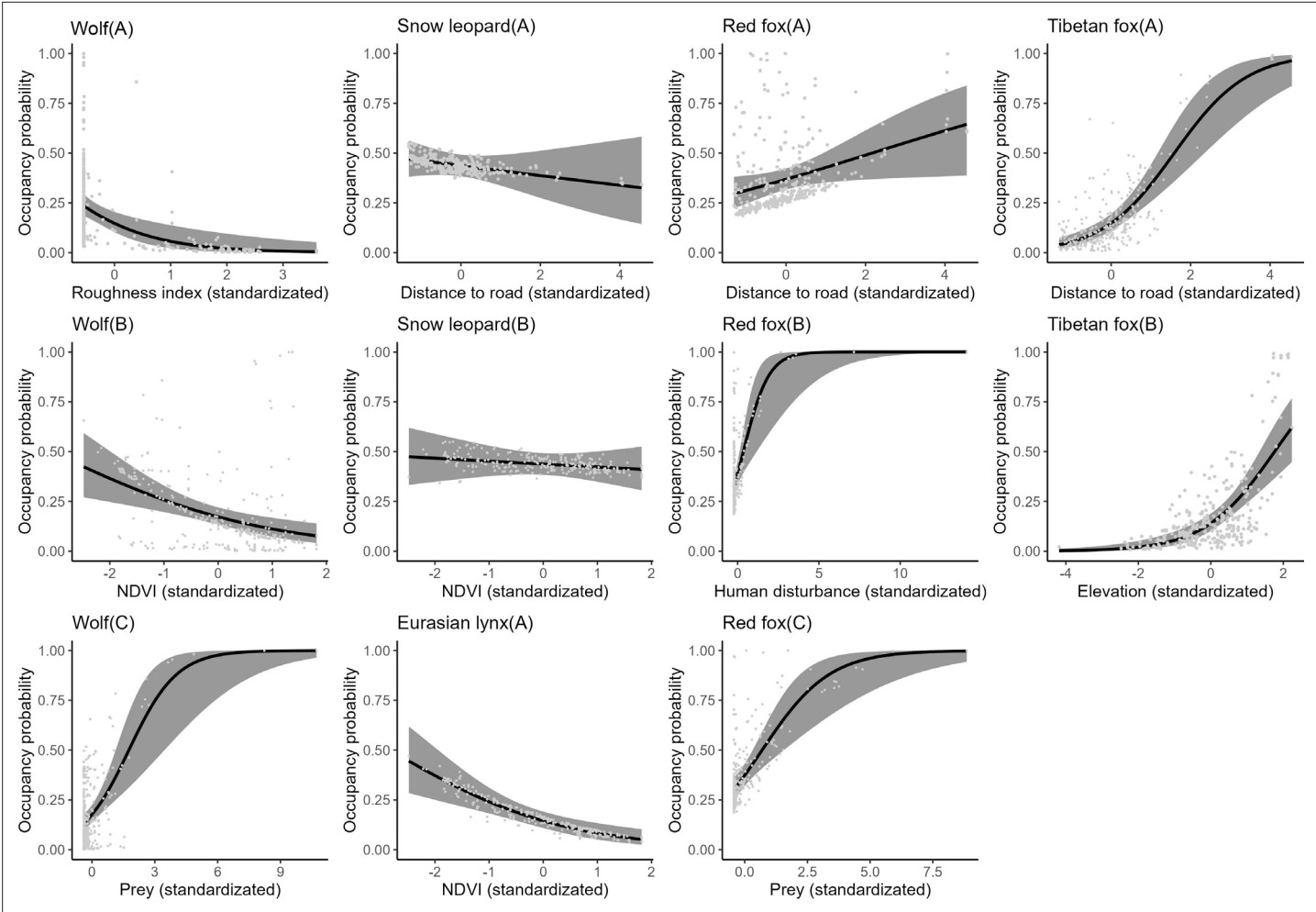

**Figure 1.** The relationship between carnivore species occupancy probability and covariates by the optimal models (ΔAIC≤2). The solid line represents the fitted polynomial regression and the gray area represents 95% confidence intervals.

the most pronounced of the three. Apex carnivore species tended to overlap temporally or trophically, mesocarnivore species had high dietary overlap with each other, and apex carnivore and mesocarnivore species displayed similarity in temporal use. Pika, blue sheep, and livestock were found to make up a large proportion of carnivore diet. Resource competition between wolf and snow leopard and the interspecific competition between Pallas's cat and Tibetan fox were strong in this area. Based on the findings presented above, we recommend targeted efforts for enhanced protection and management in several key areas. First, greater efforts are needed to protect habitat, including the establishment of habitat corridors and optimizing grassland fence layouts to safeguard migration routes. Second, resource competition should be carefully monitored between snow leopards and wolves, as well as between Pallas's cat and Tibetan foxes. More attention is needed for pika at our study site considering the role they play in the conservation of Pallas's cat and Tibetan fox populations. Previous poisoning campaigns targeting pikas were initiated out of concern for grassland degradation (**Smith and Foggin, 1999**). Recent research underscores the importance of pika population health and habitat for the distribution of Pallas's cats (**Greenspan and Giordano, 2021**). It's crucial to note that poisoning campaigns targeting small mammals may be incredibly dangerous for mesocarnivores feeding on them due to secondary poisoning, necessitating caution in such conservation strategies. Further, vigilance regarding the preponderance of smaller prey in predator diets is vital, as this may indicate severe loss of larger prey, which will increase the risk of interference competition (**Steinmetz et al., 2021**). Third, efforts should focus on restoring vulnerable wild prey populations, strengthening grazing area management, and supporting livelihoods of herders to mitigate livestock predation

**Table 3.** Spatial overlap (Sørensen's index), diel activity overlap (Δ), and dietary overlap (Pianka's index), as well as confidence intervals for carnivore species.

| | Sørensen's index | Δ | Pianka's index |
|---|---|---|---|
| Wolf – Snow leopard | 0.277 | 0.676 (0.562–0.756) | 0.892 (0.804–0.982) |
| Wolf – Eurasian lynx | 0.272 | 0.661 (0.541–0.759) | 0.585 (0.141–0.881) |
| Snow leopard – Eurasian lynx | 0.305 | 0.900 (0.854–0.992) | 0.458 (0.160–0.886) |
| Wolf – Pallas's cat | – | – | 0.658 (0.053–0.950) |
| Wolf – Red fox | 0.365 | 0.497 (0.359–0.563) | 0.811 (0.497–0.962) |
| Wolf – Tibetan fox | 0.350 | 0.855 (0.777–0.937) | 0.689 (0.456–0.967) |
| Snow leopard – Pallas's cat | – | – | 0.354 (0.092–0.827) |
| Snow leopard – Red fox | 0.477 | 0.814 (0.756–0.863) | 0.586 (0.568–0.941) |
| Snow leopard – Tibetan fox | 0.292 | 0.711 (0.629–0.744) | 0.390 (0.299–0.905) |
| Eurasian lynx – Pallas's cat | – | – | 0.376 (0–0.921) |
| Eurasian lynx – Red fox | 0.205 | 0.800 (0.728–0.878) | 0.536 (0.442–0.910) |
| Eurasian lynx – Tibetan fox | 0.198 | 0.695 (0.601–0.756) | 0.385 (0.078–0.919) |
| Pallas's cat – Red fox | – | – | 0.945 (0.369–0.988) |
| Pallas's cat – Tibetan fox | – | – | 0.997 (0–1) |
| Red fox – Tibetan fox | 0.349 | 0.576 (0.467–0.580) | 0.949 (0.279–0.988) |

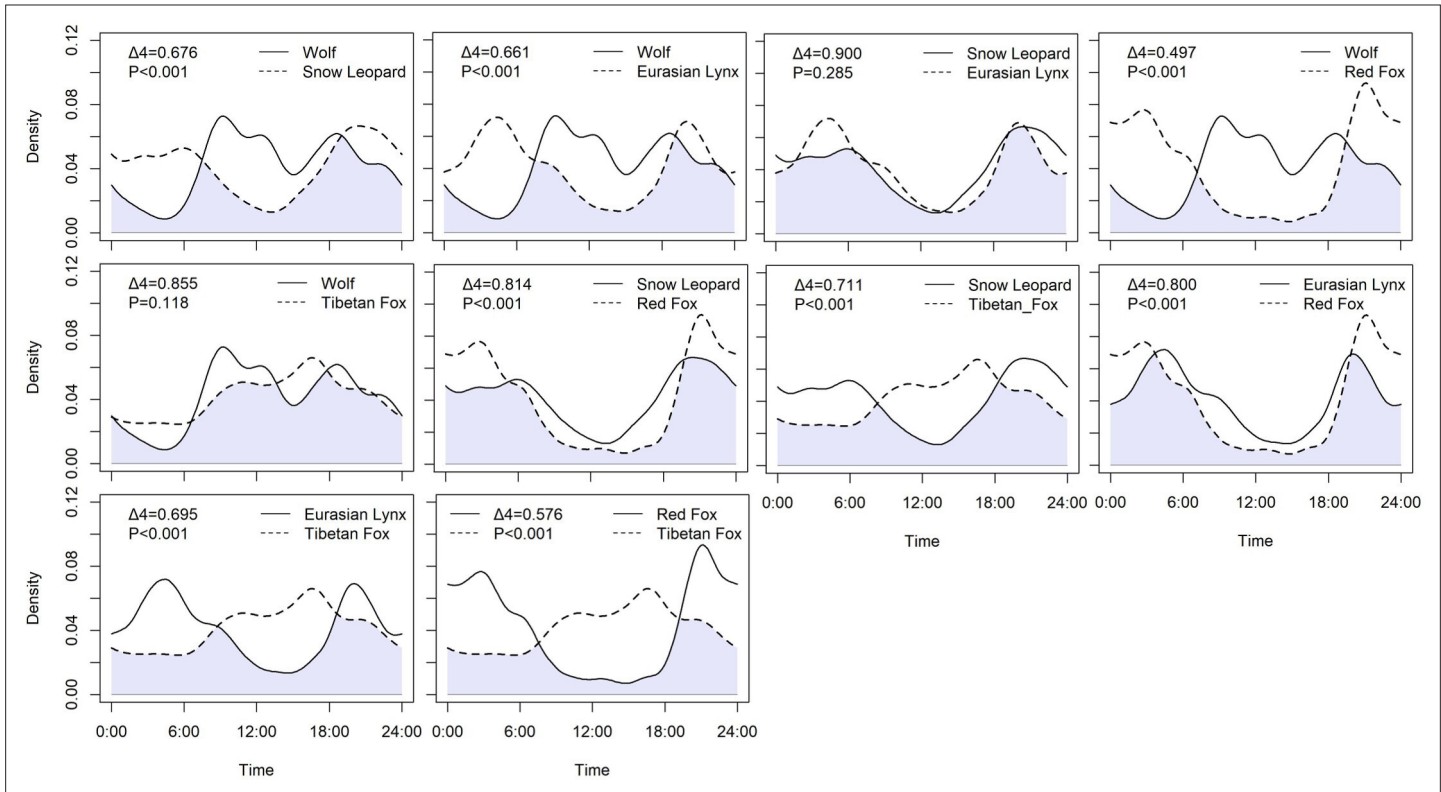

**Figure 2.** Daily activity patterns of carnivore species. Shades of lavender indicate temporal areas of overlap. The p-values are derived from Wald tests. A p-value less than 0.05 indicates that the parameter significantly contributes to the model.

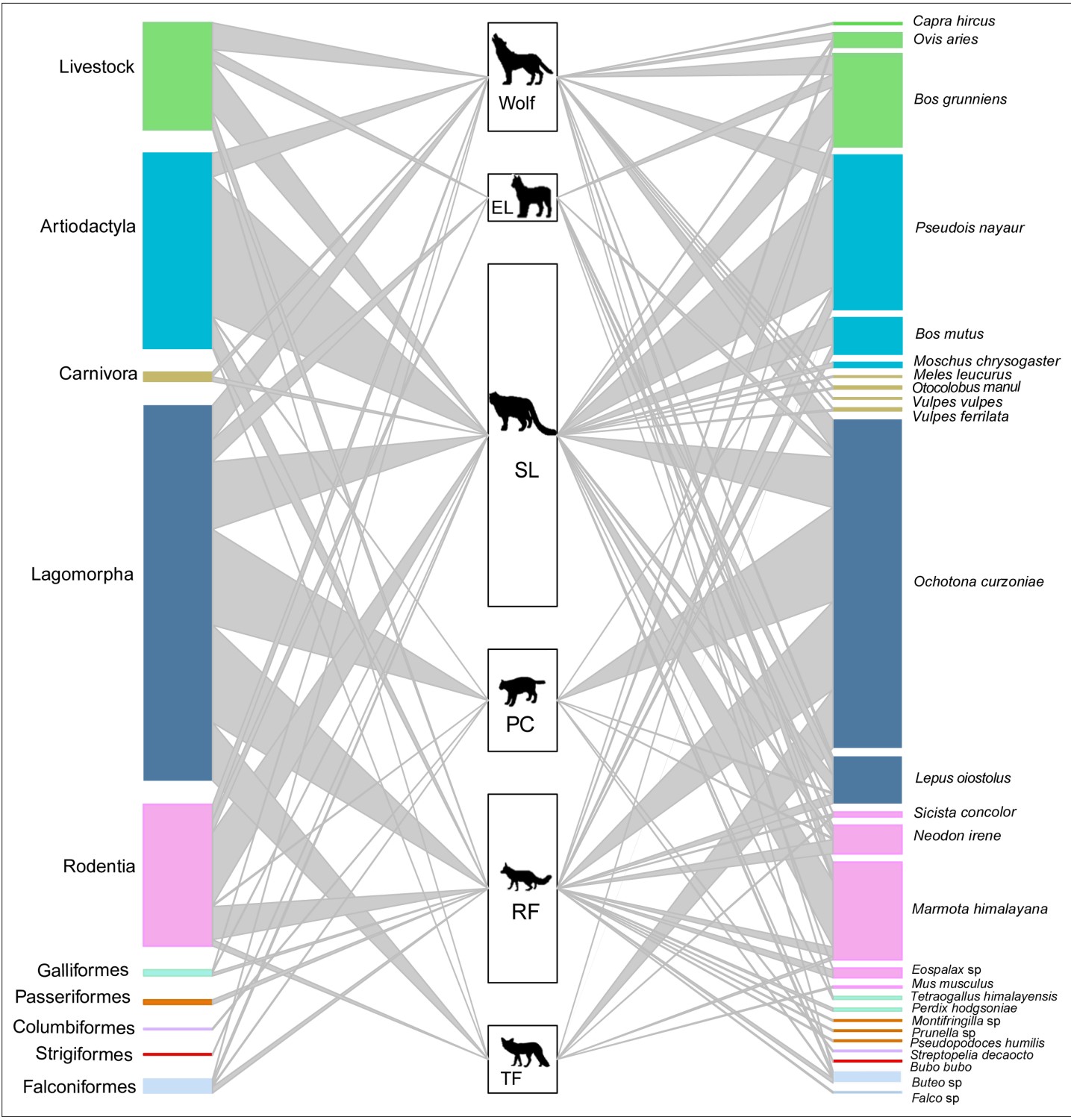

**Figure 3.** The food web of carnivore species (SL – snow leopard, EL – Eurasian lynx, PC – Pallas's cat, RF – red fox, TF – Tibetan fox). The heights of the left bars represent the frequency of occurrence of the taxonomic order of prey species, the middle bars represent the number of samples for each carnivore, and the heights of the right bars represent the frequency of occurrence of prey species in scats. The colors of prey match the taxonomic orders. The connecting line widths represent the prey frequency of occurrence in the diet of each carnivore species.

The online version of this article includes the following figure supplement(s) for figure 3:

**Figure supplement 1.** The number of scats belonging to each host species among 480 scats samples.

**Figure supplement 2.** The frequency of occurrence (FOO) percentage of each prey order by carnivore.

and consequent human-wildlife conflicts. Regular patrols of protected areas and core habitats are essential, along with educational outreach to conservation staff and herders. Our study corroborates and complements the findings of prior studies on these species and their coexistence mechanisms, offering insights crucial for wildlife conservation in the region.

## Materials and methods

### Study sites

The Qilian Mountains laterally span Gansu and Qinghai Provinces in China, located on the northeastern edge of the Qinghai-Tibetan Plateau (*Figure 4*). Qilian Mountain National Park covers an area of approximately 52,000 km², with an average elevation of over 3000 m. The area is an alpine ecosystem with a typical plateau continental climate. The average annual temperature is below –4°C and the average annual rainfall is about 400 mm, with habitats mainly consisting of deserts, grassland, meadows, and wetland (*Zheng, 2011*). Wildlife present include the wolf (*Canis lupus*), snow leopard (*Panthera uncia*), Eurasian lynx (*Lynx lynx*), red fox (*Vulpes vulpes*), Tibetan fox (*Vulpes ferrilata*), Tibetan brown bear (*Ursus arctos*), Chinese mountain cat (*Felis bieti*), wild yak (*Bos mutus*), blue sheep (*Pseudois nayaur*), alpine musk-deer (*Moschus chrysogaster*), Tibetan antelope (*Pantholops hodgsonii*), Himalayan marmot (*M. himalayana*), woolly hare (*L. oiostolus*), and plateau pika (*Ochotona curzoniae*), among others (*Ma et al., 2021*; *Xue et al., 2019*).

### Camera trap monitoring and noninvasive sampling

The study area was subdivided into sample squares of 25 km² (5 km×5 km) as a geographical reference for placing camera survey sites and collecting scat samples (*Xue et al., 2019*). Species occurrence was recorded using camera trap monitoring (Model Ltl-6210; Shenzhen Ltl Acorn Electronics Co. Ltd., Shenzhen, China). Two camera traps were placed in each square with a distance of at least 1 km between them. However, due to limitations of terrain, landform, road accessibility, and other factors, the number of camera trap in some squares was adjusted in accordance with field conditions.

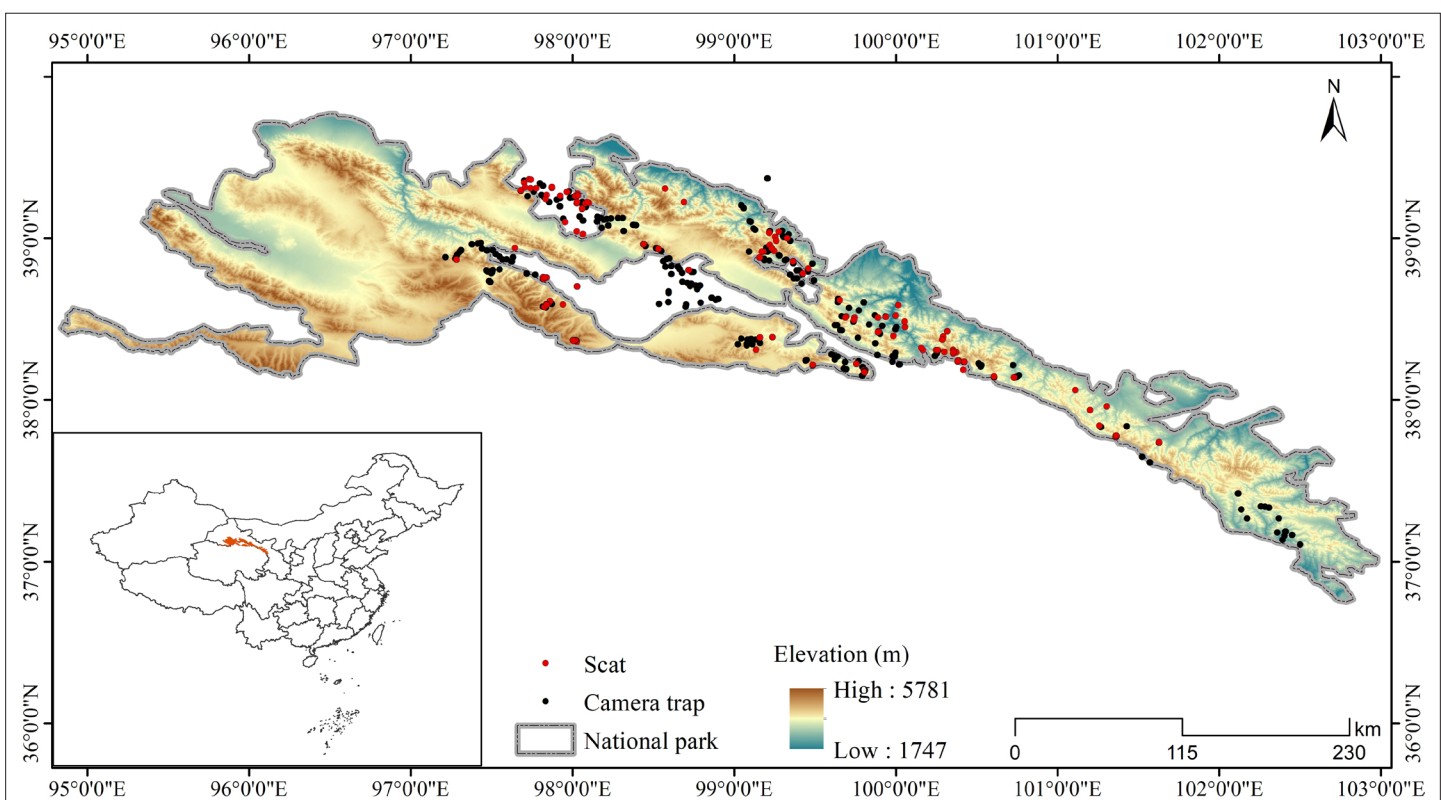

**Figure 4.** Locations of camera trap stations and scat collection sites in this study.

Camera traps were set in areas believed to be important to and heavily used by wildlife, such as the bottoms of cliffs, sides of boulders, valleys, and ridges along movement corridors. Taking into account the fact that mammalian communities are sensitive to seasonality, we used camera traps to monitor animals with an extensive survey effort from December 2016 to February 2022, covering the activity of animal species in different seasons, which can reflect the overall distribution of carnivores. We placed a total of 280 infrared cameras at the study site (*Figure 4*), set them to be active for 4–6 months, and considered possible relocation to another position based on animal detection in an effort to improve estimates of the occupancy and detection rates for both common and rare species (*Kays et al., 2020*). The camera trap was set to record the time and date on a 24 hr clock when triggered, and to record a 15 s video and 1 photo with an interval of 2 min between any two consecutive triggers. The sum of camera trap effective days was defined by the total amount of trapping effort during the sampling period, which was calculated from the time the camera was placed in operation to the time the last video or photograph was taken. We visited each camera two to three times a year to download photos and check batteries.

Noninvasive sampling of scats believed to be of carnivore origin were collected within camera trapping areas. A small portion of scat (approximately 1/3) was broken off and stored in a 15 ml centrifuge tubes with silica desiccant covered by clean filter paper to separate the desiccant from the scat (*Janečka et al., 2008*). Gloves were replaced between sampling to avoid cross-contamination. Sampling place, date, and sample number were labeled on the tube; GPS coordinates, elevation, and nearby landscape features were recorded on the sample collection sheet (*Hacker et al., 2021*). A total 480 scat samples were collected from April 2019 to June 2021 (*Figure 4*).

## Data analysis
### Spatial analysis
To investigate the spatial distribution of carnivores, as well as the influence of environmental factors on the site occupancy of species in the study area, we performed single-season, single-species occupancy models to estimate carnivores' occupancy ($\psi$) and detection (p) probability (*Li et al., 2022b*; *MacKenzie, 2018*; *Moreno-Sosa et al., 2022*). To ensure capture independence, only photo or video records at intervals greater than 30 min for same species were included in the data analysis (*Li et al., 2010*). We created a matrix recording whether each carnivore species was detected (1) or not (0) across several 30-day intervals (i.e. 0–30, 31–60, 61–90, 91–120, 121–150, >150 days) for each camera location. Based on the previous studies of habitat use of carnivores (*Greenspan and Giordano, 2021*; *Alexander et al., 2016*; *Gorczynski et al., 2022*), we selected terrain, vegetation, biological factors, and disturbance to construct the model. Terrain is a fundamental element of wildlife habitat and closely linked to other environmental factors (*Chen et al., 2024*). Terrain variables include elevation (ele) and roughness index (rix). Vegetation variables include ndvi and provide information on the level of habitat concealment. Biological variables include prey abundance (the number of independent photos of their preferred prey based on dietary analysis in this study, wolf and snow leopard: artiodactyla including livestock; Eurasian lynx and Pallas's cat: lagomorpha; red fox and Tibetan fox: lagomorpha and rodentia) and reflect habitat preference and distribution patterns of carnivores. Disturbance variables include distance to roads (disrd) and human disturbances (hdis, the number of independent photos of herdsman and livestock) and can provide insight into the habitat selection and behavior patterns of carnivores. In addition, we used elevation, human disturbance, and prey as covariates that affect detection rate. Road data were obtained from Open Street Map (OSM, https://www.openstreetmap.org). Others environmental data were obtained from the Resource and Environment Science and Data Center (https://www.resdc.cn). We fitted all possible combinations of covariates and used Akaike's information criterion (AIC) to rank candidate models, and selected $\Delta$AIC$\leq$2 model as the optimal model. If more than one optimal model resulted, then covariate estimates were obtained by using the equal-weight average.

Carnivore co-occurrence was evaluated using the Sørensen similarity index (value = 0, indicating maximum segregation and value = 1, indicating maximum co-occurrence) based on binary presence-absence data within the 5 km × 5 km camera trap grid (*Torretta et al., 2021*; *Sorensen, 1948*). Spatial analyses were performed using ArcGIS 10.8 (ESRI Inc), the 'vegan' packages (*Oksanen et al., 2019*), and 'unmarked' package (*Fiske and Chandler, 2011*) for R studio (version 1.1.463).

## Temporal analysis

Estimates of the coefficient of overlap (Δ) for activity patterns were estimated using the non-parametric kernel density method and applying time data obtained by the camera traps. Because the smallest sample had more than 50 records, we used the $\Delta_4$ estimator for pairwise comparisons between carnivore activity patterns, and used a smooth bootstrap scheme to generate 1000 resamples with 95% confidence intervals to test the reliability of the overlap value (*Ridout and Linkie, 2009*). Activity pattern analyses were performed using the 'overlap' R package. Values of the $\Delta_4$ estimates were calculated relative to 1000 simulated null models of randomized overlap data using the 'compareCkern' function in the 'activity' R package to test for differences in daily activity patterns (*Ridout and Linkie, 2009*; *Rowcliffe et al., 2014*).

## Species identification and dietary analysis

Host species and diet were identified using metabarcoding. DNA was extracted using the QIAamp Fast DNA Stool Mini Kit (QIAGEN, Hilden, Germany) following standard protocols and the MT-RNR1 (12S) and COX1 (cytochrome *c* oxidase subunit I) gene segments amplified using 12SV5-F/R primer and COX1 primers, respectively (*Hacker et al., 2021*; *Riaz et al., 2011*). PCR conditions followed the methods described in *Hacker et al., 2021*. The resulting library was sequenced on an Illumina NovaSeq platform and 250 bp paired-end reads were generated (Guangdong Magigene Biotechnology Co., Ltd., Guangzhou, China).

We used CLC Genomics Workbench version 12.0 to determine the host species as well as the prey consumed by each carnivore by mapping sequence reads to reference sequences of possible prey downloaded from GenBank and BOLD (Barcode of Life Data Systems) with representative haplotypes compiled into one.fasta file. Raw reads were required to have at least 98% similarity across at least 90% of the reference sequence for mapping (*Hacker et al., 2021*). Species and prey identification were made based on the reference taxa with the highest number of reads mapped and the fewest mismatches. Samples in which species could not be identified were analyzed to ensure the reference file was not incomplete by using the de nova assembly tool in CLC, then blasting the resulting contig sequence with the nucleotide databases in NCBI (https://blast.ncbi.nlm.nih.gov/Blast.cgi). As an additional precaution, the geographical range of the determined host and prey species was researched using the IUCN Red List (https://www.iucnredlist.org/) to ensure that it overlapped with the study site. For complete methods on data parameters and methods used, see *Hacker et al., 2021*.

Dietary data were summarized by the frequency of occurrence of prey species in scats observed. The 'bipartite' R package was used to construct food web networks (*Dormann, 2011*). Dietary diversity for each carnivore host species was assessed by calculating richness and Shannon's index (*Shannon and Weaver, 1949*). Interspecific dietary niche overlap was evaluated using Pianka's index ($O_{jk}$) (value = 0, no dietary overlap and value = 1, complete dietary overlap) and 95% confidence intervals were obtained by bootstrapping with 1000 resamples via the 'spaa' R package (*Zhang, 2016*). Dietary similarity between any two given carnivore species was assessed by calculating the inversed value of Jaccard's index based on binary presence-absence data of prey.

## Acknowledgements

We would like to thank Mr. Jiong He, Yayue Gao, Duifang Ma, Liji Wu, Dazhi Hu, and other colleagues of Qilianshan National Park for their generous assistance in the field surveys. We thank Dr. Charlotte Hacker for editing the English text of this manuscript.

## Additional information

### Funding

| Funder | Grant reference number | Author |
| --- | --- | --- |
| National Natural Science Foundation of China | 32201430 | Jia Li |

| Funder | Grant reference number | Author |
|---|---|---|
| National Natural Science Foundation of China | 32101409 | Yu Zhang |
| Welfare Project of the National Scientific Research Institution | CAFYBB2019ZE003 | Yuguang Zhang |

The funders had no role in study design, data collection and interpretation, or the decision to submit the work for publication.

## Author contributions

Wei Cong, Investigation, Writing – original draft, Writing – review and editing; Jia Li, Funding acquisition, Investigation, Writing – review and editing; Charlotte Hacker, Investigation, Methodology, Writing – review and editing; Ye Li, Lixiao Jin, Yi Zhang, Investigation; Yu Zhang, Funding acquisition, Investigation; Diqiang Li, Funding acquisition, Methodology, Project administration; Yadong Xue, Data curation, Investigation, Project administration, Writing – review and editing; Yuguang Zhang, Data curation, Funding acquisition, Investigation, Methodology, Project administration, Writing – review and editing

## Author ORCIDs

Wei Cong ⓘ https://orcid.org/0009-0005-6071-9564
Yuguang Zhang ⓘ http://orcid.org/0000-0001-9801-8556

Reviewer #1 (Public Review): https://doi.org/10.7554/eLife.90559.3.sa1
Reviewer #2 (Public Review): https://doi.org/10.7554/eLife.90559.3.sa2
Author response https://doi.org/10.7554/eLife.90559.3.sa3

# Additional files

## Supplementary files

• Supplementary file 1. Dietary composition, diversity, as well as similarity among six carnivore species. (a) The frequency of occurrence (FOO) of prey found in carnivore diet. (b) Dietary diversity indices for each carnivore species. (c) Jaccard distance for prey items in diets using binary presence-absence data.

• MDAR checklist

## Data availability

All data generated or analysed during this study are included in the manuscript and supporting files.

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
