## [Editor Report · eLife assessment]

This paper provides an **important** assessment of competition dynamics allowing coexistence of the carnivore guild within a large national park in China. Multiple surveying techniques (camera traps and DNA metabarcoding) provide **convincing** evidence that spatial segregation represents the main strategy of coexistence, while species have a certain degree of temporal and dietary overlap. Altogether, the manuscript provides information critical to the conservation and management agenda of the park.

---

## [Referee Report · Reviewer #1 (Public Review)]

Summary:

This work explored intra and interspecific niche partitioning along spatial, temporal, and dietary niche partitioning between apex carnivores and mesocarnivores in the Qilian Mountain National Park of China, using camera trapping data and DNA metabarcoding sequencing data. They conclude that spatial niche partitioning plays a key role in facilitating the coexistence of apex carnivore species, spatial and temporal niche partitioning facilitate the coexistence of mesocarnivore species, and spatial and dietary niche partitioning facilitate the coexistence between apex and mesocarnivore species. The information presented in this study is important for wildlife conservation and will contribute substantially to the current understanding of carnivore guilds and effective conservation management in fragile alpine ecosystems.

Strengths:

Extensive fieldwork is evident in the study. Aiming to cover a large percentage of the Qilian Mountain National Park, the study area was subdivided into squares, as a geographical reference to distribute the sampling points where the camera traps were placed and the excreta samples were collected.

They were able to obtain many records in their camera traps and collected many samples of excreta. This diversity of data allowed them to conduct robust analyses. The data analyses carried out were adequate to obtain clear and meaningful results that enabled them to answer the research questions posed. The conclusions of this paper are mostly well supported by data.

The study has demonstrated the coexistence of carnivore species in the landscapes of the Qilian Mountains National Park, complementing the findings of previous studies. The information presented in this study is important for wildlife conservation and will contribute substantially to the current understanding of carnivore guilds and effective conservation management in fragile alpine ecosystems.

Weaknesses:

It is necessary to better explain the methodology because it is not clear what is the total sampling effort. In methodology, they only claim to have used 280 camera traps, and in the results, they mention that there are 319 sampling sites. However, the total sampling effort (e.g. total time of active camera traps) carried out in the study and at each site is not specified.

---

## [Referee Report · Reviewer #2 (Public Review)]

Summary:

The study entitled "Different coexistence patterns between apex carnivores and mesocarnivores based on temporal, spatial, and dietary niche partitioning analysis in Qilian Mountain National Park, China" by Cong et al. addresses the compelling topic of carnivores' coexistence in a biodiversity hotspot in China. The study is interesting given it considers all three components affecting sympatric carnivores' distribution and co-occurrence, namely the temporal, the spatial, and the dietary partition within the carnivore guild. The authors have found that spatial co-occurrence is generally low, which represents the major strategy for coexistence, while there is temporal and dietary overlap. I also appreciated the huge sampling effort carried out for this study by the authors: they were able to deploy 280 camera trapping sites (which became 322 in the result section?) and collect a total of 480 scat samples. However, I have some concerns about the study on the non-consideration of the human dimension and potential anthropogenic disturbance that could affect the spatial and temporal distribution of carnivores, the choice of the statistical model to test co-occurrence, and the lack of clearly stated ecological hypotheses.

Strengths:

The strengths of the study are the investigation of all three major strategies that can mitigate carnivores' coexistence, therefore, the use of multiple monitoring techniques (both camera trapping and DNA metabarcoding) and the big dataset produced that consists of a very large sampled area with a noteworthy number of camera tap stations and many scat samples for each species.

Weaknesses:

I think that some parts of the manuscript should be written better and more clearly. A clear statement of the ecological hypotheses that could affect the partitioning among the carnivore guild is lacking. I think that the human component (thus anthropogenic disturbance) should have been considered more in the spatial analyses given it can influence the use of the environment by some carnivores. Additionally, a multi-species co-occurrence model would have been a more robust approach to test for spatial co-occurrence given it also considers imperfect detection.

Temporal and dietary results are solid and this latter in particular highlights a big predation pressure on some prey species such as the pika. This implies important conservation and management implications for this species, and therefore for the trophic chain, given that (i) the pika population should be conserved and (ii) a potential poisoning campaign against small mammals could be incredibly dangerous also for mesocarnivores feeding on them due to secondary poisoning.

---

## [Author Response]

The following is the authors’ response to the original reviews.

**Public Reviews:**

**Reviewer #1 (Public Review):**
Summary:This work explored intra and interspecific niche partitioning along spatial, temporal, and dietary niche partitioning between apex carnivores and mesocarnivores in the Qilian Mountain National Park of China, using camera trapping data and DNA metabarcoding sequencing data. They conclude that spatial niche partitioning plays a key role in facilitating the coexistence of apex carnivore species, spatial and temporal niche partitioning facilitate the coexistence of mesocarnivore species, and spatial and dietary niche partitioning facilitate the coexistence between apex and mesocarnivore species. The information presented in this study is important for wildlife conservation and will contribute substantially to the current understanding of carnivore guilds and effective conservation management in fragile alpine ecosystems.Strengths:Extensive fieldwork is evident in the study. Aiming to cover a large percentage of the Qilian Mountain National Park, the study area was subdivided into squares, as a geographical reference to distribute the sampling points where the camera traps were placed and the excreta samples were collected.They were able to obtain many records in their camera traps and collected many samples of excreta. This diversity of data allowed them to conduct robust analyses. The data analyses carried out were adequate to obtain clear and meaningful results that enabled them to answer the research questions posed. The conclusions of this paper are mostly well supported by data.The study has demonstrated the coexistence of carnivore species in the landscapes of the Qilian Mountains National Park, complementing the findings of previous studies. The information presented in this study is important for wildlife conservation and will contribute substantially to the current understanding of carnivore guilds and effective conservation management in fragile alpine ecosystems.Weaknesses:It is necessary to better explain the methodology because it is not clear what is the total sampling effort. In methodology, they only claim to have used 280 camera traps, and in the results, they mention that there are 319 sampling sites. However, the total sampling effort (e.g. total time of active camera traps) carried out in the study and at each site is not specified.

Thanks a lot for this detailed review! We apologize for not offering a distinct description of the overall sampling effort. In this study, we deployed 280 camera trappings, and these cameras were active for approximately 4 to 6 months. We visited each camera 2 to 3 times annually to download photos and check the batteries. In case some cameras failed to capture the targeted carnivore, we would relocate the positions of those cameras. Eventually, we collected 322 camera trapping sites, among which 3 cameras malfunctioned due to loss. As a result, we analyzed data from 319 camera sites and obtained 14,316 independent detections over 37,192 trap-days.

We have added this information as follows in lines 132 to lines 143: “Taking into account the fact that mammalian communities are sensitive to seasonality, we used camera traps to monitor animals with an extensive survey effort from December 2016 to February 2022, covering the activity of animal species in different seasons, which can reflect the overall distribution of carnivores. We placed a total of 280 infrared cameras at the study site, set them to be active for 4 to 6 months, and considered possible relocation to another position based on animal detection in an effort to improve estimates of the occupancy and detection rates for both common and rare species (Figure 1) (Kays et al., 2020). The camera trap was set to record the time and date on a 24 hr clock when triggered, and to record a 15s video and 1 photo with an interval of 2 minutes between any two consecutive triggers. The sum of camera trap effective days was defined by the total amount of trapping effort during the sampling period, which was calculated from the time the camera was placed in operation to the time the last video or photograph was taken. We visited each camera 2 to 3 times a year to download photos and check batteries.” and lines 228 to lines 232: “A total of 322 camera trap sites were surveyed after relocating infrared cameras that did not capture any target carnivore species. A total of 3 cameras were considered to have failed due to loss. We analyzed data from 319 camera sites and obtained 14,316 independent detections during a total effort of 37,192 effective camera trap days. We recorded wolf in 26 sites, snow leopard in 109 sites, Eurasian lynx in 36 sites, red fox in 92 sites, and Tibetan fox in 34 sites.”

**Reviewer #2 (Public Review):**
Summary:The study entitled "Different coexistence patterns between apex carnivores and mesocarnivores based on temporal, spatial, and dietary niche partitioning analysis in Qilian Mountain National Park, China" by Cong et al. addresses the compelling topic of carnivores' coexistence in a biodiversity hotspot in China. The study is interesting given it considers all three components affecting sympatric carnivores' distribution and co-occurrence, namely the temporal, the spatial, and the dietary partition within the carnivore guild. The authors have found that spatial co-occurrence is generally low, which represents the major strategy for coexistence, while there is temporal and dietary overlap. I also appreciated the huge sampling effort carried out for this study by the authors: they were able to deploy 280 camera trapping sites (which became 322 in the result section?) and collect a total of 480 scat samples. However, I have some concerns about the study on the non-consideration of the human dimension and potential anthropogenic disturbance that could affect the spatial and temporal distribution of carnivores, the choice of the statistical model to test co-occurrence, and the lack of clearly stated ecological hypotheses.Strengths:The strengths of the study are the investigation of all three major strategies that can mitigate carnivores' coexistence, therefore, the use of multiple monitoring techniques (both camera trapping and DNA metabarcoding) and the big dataset produced that consists of a very large sampled area with a noteworthy number of camera trap stations and many scat samples for each species.Weaknesses:I think that some parts of the manuscript should be written better and more clearly. A clear statement of the ecological hypotheses that could affect the partitioning among the carnivore guild is lacking. I think that the human component (thus anthropogenic disturbance) should have been considered more in the spatial analyses given it can influence the use of the environment by some carnivores. Additionally, a multi-species co-occurrence model would have been a more robust approach to test for spatial co-occurrence given it also considers imperfect detection.

Thank you very much for your valuable comments and suggestions. We checked and edited the manuscript, and we thought the English level was improved.

(1) According to your suggestion, we added the competitive exclusion and niche differentiation hypothesis with space, time and diets axis to explain co-occurrence relationship among species in the introduction as follow: “The competitive exclusion principle dictates that species with similar ecological requirements are unable to successfully coexist (Hardin, 1960; Gause, 1934). Thus, carnivores within a guild occupy different ecological niches based on a combination of three niche dimensions, i.e. spatial, temporal, and trophic (Schoener, 1974). Spatially, carnivore species within the same geographic area exhibit distinct distributions that minimize overlap in resource use and competition. For example, carnivores can partition habitats based on habitat feature preferences and availability of prey (De Satgé et al., 2017; Garrote and Pérez De Ayala, 2019; Gołdyn et al., 2003; Strampelli et al., 2023). Temporally, differences in seasonal or daily activity patterns among sympatric carnivores can reduce competitive interactions and facilitate coexistence. For example, carnivores can exhibit temporal segregation in their foraging behaviors, such as diurnal versus nocturnal activity, to avoid direct competition (Finnegan et al., 2021; Nasanbat et al., 2021; Searle et al., 2021). Trophically, carnivore species can diversify their diets to exploit different prey species or sizes, thereby reducing competition for food resources. For example, carnivores can exhibit dietary specialization to optimize their foraging efficiency and minimize competitive pressures (Steinmetz et al., 2021).”

(2) In addition to distance from roads, we included human dimension as covariates influencing occupancy rates based on the number of independent photos or videos of herders and livestock detected by infrared cameras (named human disturbance and is represented by hdis). According to the results of occupancy models, we found red fox occupancy probability displayed a significant positive relationship with hdis. Moreover, the detection probability of snow leopard and Eurasian lynx decreased with increasing hdis.

We have incorporated these results into the Results as follow: “According to the findings derived from single-season, single-species occupancy models, the snow leopard demonstrated a notably higher probability of occupancy compared to other carnivore species, estimated at 0.437 (Table 1). Conversely, the Eurasian lynx exhibited a lower occupancy probability, estimated at 0.161. Further analysis revealed that the occupancy probabilities of the wolf and Eurasian lynx declined with increasing Normalized Difference Vegetation Index (NDVI) (Table 2, Figure 2). Additionally, wolf occupancy probability displayed a negative relationship with roughness index and a positive relationship with prey availability. Snow leopard occupancy probabilities exhibited a negative relationship with distance to roads and NDVI. In contrast, both red fox and Tibetan fox demonstrated a positive relationship with distance to roads. Moreover, red fox occupancy probability increased with higher human disturbance and greater prey availability. The detection probabilities of wolf, snow leopard, red fox, and Tibetan fox exhibited an increase with elevation (Table 2). Moreover, there was a positive relationship between the detection probability of Tibetan fox and prey availability. The detection probabilities of snow leopard and Eurasian lynx declined as human disturbance increased.”

(3) We appreciate the suggestion to use a multi-species co-occurrence model to test spatial co-occurrence. We attempted a multispecies occupancy modeling to analysis the five species in our study followed the method of Rota et al. (2016). Initially, we simplified the candidate models by adopting a single-season, single-species occupancy model. We selected occupancy covariates from the best model as the best covariates for each species and used them to establish multispecies occupancy models. Unfortunately, the final model results did not converge. We are investigating potential solutions to resolve this problem.

Rota CT, Ferreira MAR, Kays RW, Forrester TD, Kalies EL, McShea WJ, Parsons AW, Millspaugh JJ. 2016. A multispecies occupancy model for two or more interacting species. Methods Ecol Evol **7**:1164–1173. doi:10.1111/2041-210X.12587

Temporal and dietary results are solid and this latter in particular highlights a big predation pressure on some prey species such as the pika. This implies important conservation and management implications for this species, and therefore for the trophic chain, given that (i) the pika population should be conserved and (ii) a potential poisoning campaign against small mammals could be incredibly dangerous also for mesocarnivores feeding on them due to secondary poisoning.

Thank you for your thoughtful comments. We appreciate your recognition of the temporal and dietary findings, particularly the highlighted predation pressure on prey species like the pika. These observations indeed underscore critical implications for conservation and management. The necessity to conserve the pika population is paramount for its role in maintaining the stability of the trophic chain within its ecosystem. As you rightly pointed out, any disruption to this delicate balance, including through predation or indirect threats like poisoning campaigns, could have far-reaching consequences. Regarding the potential risks associated with poisoning campaigns targeting small mammals, we acknowledge the significant concerns raised about secondary poisoning affecting mesocarnivores. This underscores the need for careful consideration in pest control strategies and the adoption of measures that minimize unintended ecological impacts. Our findings suggest several practical implications for conservation and management. Conservation efforts should focus on vulnerable prey populations such as the pika, while management strategies could include regulatory frameworks and community education to mitigate risks associated with pest control methods. We believe our study contributes valuable insights into the complexities of predator-prey dynamics and the broader implications for ecosystem health. By integrating these findings into conservation practices, we can work towards ensuring the sustainability of natural systems and the species that depend on them.

**Reviewer #1 (Recommendations For The Authors):**
To better explain the methodology and the sampling effort I recommend reviewing e.g. Kays et al. 2020. An empirical evaluation of camera trap study design: How many, how long, and when?. Methods in Ecology and Evolution, 11(6), 700-713. https://besjournals.onlinelibrary.wiley.com/doi/epdf/10.1111/2041-210X.13370.

Thank you for this valuable suggestion! According to this reference, we have added this information to explain the methodology and the sampling effort as follow: “Taking into account the fact that mammalian communities are sensitive to seasonality, we used camera traps to monitor animals with an extensive survey effort from December 2016 to February 2022, covering the activity of animal species in different seasons, which can reflect the overall distribution of carnivores. We placed a total of 280 infrared cameras at the study site, set them to be active for 4 to 6 months, and considered possible relocation to another position based on animal detection in an effort to improve estimates of the occupancy and detection rates for both common and rare species (Figure 1) (Kays et al., 2020). The camera trap was set to record the time and date on a 24 hr clock when triggered, and to record a 15s video and 1 photo with an interval of 2 minutes between any two consecutive triggers. The sum of camera trap effective days was defined by the total amount of trapping effort during the sampling period, which was calculated from the time the camera was placed in operation to the time the last video or photograph was taken. We visited each camera 2 to 3 times a year to download photos and check batteries.”

**Reviewer #2 (Recommendations For The Authors):**
I have some concerns about the manuscript.I find that the manuscript should be written more clearly: some sentences are not straightforward to understand given the presence of structural errors that make the text hard to read; the paragraphs should be written in a more harmonic way (without logical leaps) with a smoother change of topic between paragraphs, especially in the introduction.

We appreciate your constructive comments, which have helped us improve the clarity and coherence of the manuscript. We have revised the introduction to provide a clearer outline of the paper's structure and objectives. Specifically, we have rephrased complex sentences and removed ambiguities to ensure that each idea is communicated more straightforwardly. We providing clearer links between ideas and avoiding abrupt shifts in topics to ensure that a smoother transition between paragraphs.

I feel like the strength of merging the two techniques (camera trapping and DNA metabarcoding) is not brought up enough, while the disadvantage of this approach is not even mentioned (e.g., the increasing costs).

Thanks a lot for this valuable comment! We have added this information to the Discussion (L356-L363) as follow: “Our study highlights the effectiveness of combining camera trapping with DNA metabarcoding for detecting and identifying both cryptic and rare species within a sympatric carnivore guild. This integrated approach allowed us to capture a more comprehensive view of species presence and interactions compared to traditional visual surveys. whereas, it is important to acknowledge the challenges associated with this technique, including the high costs of equipment and the need for specialized training and computational resources to manage and analyze the large volumes of sequence data. Despite these challenges, the benefits of this combined method in improving biodiversity assessments and understanding species coexistence outweigh the drawbacks.”

The structure of the manuscript does not follow the structure of the journal (Intro, Material and Method, Results, Discussion instead it reports the methods at the end of the main manuscript), and, most critically, I found that a clear explanation of the research hypothesis is missing: authors should clearly state they ecological hypotheses. What are your hypotheses on the co-occurrence relationship among species? What would specifically affect and change the sympatric relationships among carnivores?

Thank you for this valuable suggestion! We have revised the manuscript, that is integrated the methods section appropriately within the main body of the manuscript to ensure that it aligns with the standard sections Introduction, Materials and Methods, Results, Discussion.

We state our main ecological hypotheses concerning the co-occurrence relationships among carnivore species is based on niche differentiation hypothesis. We hypothesize that differentiation along one or more niche axes is beneficial for the coexistence of carnivorous guild in the Qilian Mountains. We expected that spatial niche differentiation promotes the coexistence of large carnivores in the Qilian Mountain region, as they are more likely than small carnivores to spatially avoid interspecific competition (Davis et al., 2018). Mesocarnivores may coexist either spatially or temporally due to increased interspecific competition for similar prey (Di Bitetti et al., 2010; Donadio and Buskirk, 2006). Nutritional niche differentiation may be a significant factor for promoting coexistence between large and mesocarnivore species due to differences in body size (Gómez-Ortiz et al., 2015; Lanszki et al., 2019). We have added ecological hypotheses in lines 101 to 110.

Another concern is that all pictures with people have been removed from the dataset, but I think that this could be a bit biased as human presence (or also the presence of livestock) could affect the spatial or temporal presence of carnivores, changing their co-occurrence dynamics. On one side, humans can be perceived as a source of disturbance by carnivores and, therefore, can cause a shift in distribution towards locations with lower human presence (or lower anthropogenic disturbance) that could further concentrate the presence of carnivores increasing the competitive interaction. Conversely, mesocarnivores could take advantage of an increasing human presence - following the human shield hypotheses - finding a refugium from larger body carnivores. From this perspective, important information on the potential anthropogenic pressure is lacking in the description of the study area: how effective is the protection effort of the park? How intense is the potential human disturbance in and around the park? Is there poaching? Intensive livestock grazing? Resources extractions? These are all factors that could affect the interactions among carnivores. Do not forget the possibility and risk of being retaliatory killed by humans due to the presence of livestock in the area. I think that incorporating the human dimension is important because it could strongly affect how carnivores perceive and use the environment. Here only the distance to the closest road has been considered. However, for example, recent research (Gorczynski et al 2022, Global Change Biology) has indeed found that co-occurrece of ecologically similar species differed in relation to increasing human density. Therefore, I think that anthropogenic disturbance is an aspect to be reckoned with and more variables as proxy of human disturbance should be considered.

Thanks a lot for this valuable comment! We acknowledge that humans can act as both a disturbance factor, potentially driving carnivores away from highly populated areas, and as a source of indirect refuge for mesocarnivores, thereby affecting competitive interactions among carnivores. We understand that poaching and resource extraction are prohibited and livestock grazing is a significant human activity within the study area. Therefore, we added human dimension as covariates influencing occupancy rates based on the number of independent photos or videos of herders and livestock detected by infrared cameras (named human disturbance and is represented by hdis). According to the results of occupancy models, we found red fox occupancy probability displayed a significant positive relationship with hdis. Moreover, the detection probability of snow leopard and Eurasian lynx decreased with increasing hdis.

In the statistical analyses section, I don't find that the statistical procedure is well described: it is not clear which occupancy model has been used (probably a single-species single-season occupancy model for each target species?), which covariates have been tested for each species and following which hypotheses. Additionally, I think that when modelling the spatial distribution of subordinate species, it should be important to include information on the spatial distribution of apex species given this could affect their occurrence on the territory. This could have been done by using the Relative Abundance Index of the apex predators as a covariate when modelling the distribution of subordinate species. Additionally, why haven't the authors used prey as a covariate for occupancy? I think that prey distribution should affect the occupancy probability more than the detection rate. Also, the authors used the Sørensen similarity index to measure associations between species. However, this association metric has been criticized (see the recent paper of Mainali et al 2022, Science Advances). I am therefore wondering: given the authors are using the occupancy framework, why don't they use a multi-species co-occurrence model that allows them to directly estimate both single-species occupancy and the co-occurrence parameter as a function of covariates (examples are Rota et al. 2016, Methods Ecol. Evol. Or Tobler et al. 2019, Ecology)? For the temporal overlap, I think that adding Figure S2 (pairwise temporal overlap) in the main text would help deliver the results of the temporal analyses more straightforwardly.

Thanks a lot for this valuable comment!

(1) The current manuscript utilizes a single-species single-season occupancy model for each target species. Additionally, we have added prey and human disturbance as occupancy covariables. We have revised the statistical analyses section to explicitly state this model choice and clarify the covariates tested for each species from lines 153 to lines170. The details are as follows: “To investigate the spatial distribution of carnivores, as well as the influence of environmental factors on the site occupancy of species in the study area, we performed single-season, single-species occupancy models to estimate carnivores’ occupancy (ψ) and detection (Pr) probability (Li et al., 2022b; MacKenzie, 2018; Moreno-Sosa et al., 2022). To ensure capture independence, only photo or video records at intervals of 30 min were was included in the data analysis (Li et al., 2020). We created a matrix recording whether each carnivore species was detected (1) or not (0) across several 30-day intervals (that is 0-30, 31-60, 61-90, 91-120, 121-150, >150 days) for each camera location. Based on the previous studies of habitat use of carnivores (Greenspan and Giordano, 2021; Alexander et al., 2016; Gorczynski et al., 2022), we selected terrain, vegetation, biological factors and disturbance to construct the model. Terrain is a fundamental element of wildlife habitat and closely linked to other environmental factors (Chen et al., 2024). Terrain variables include elevation (ele) and roughness index (rix). Vegetation variables include normalized difference vegetation index (ndvi), and provide information on the level of habitat concealment. Biological variables include prey abundance (the number of independent photos of their preferred prey based on dietary analysis in this study, wolf and snow leopard: artiodactyla including livestock; Eurasian lynx and Pallas’s cat: lagomorpha; red fox and Tibetan fox: lagomorpha and rodentia) and reflect habitat preference and distribution patterns of carnivores. Disturbance variables include distance to roads (disrd) and human disturbances (hdis, the number of independent photos of herdsman and livestock) and can provide insight into the habitat selection and behavior patterns of carnivores.”

(2) Thank you for your valuable suggestions. We acknowledge the importance of considering apex species in models of subordinate species' spatial distributions.

Nonetheless, considering the consistency of covariates for each species and the lack of interspecies interactions in single-species occupancy models, we did not include the Relative Abundance Index of the apex predators as a covariate affecting the occupancy of mesopredators. As you recommended, multi-species occupancy models that account for interspecies interactions are a robust approach. However, we attempted to use the multi-species occupancy method of Rota et al. (Rota et al., 2016), the final model results did not converge. Specifically, we selected occupancy covariates from the best model by single-species model as the best covariates for each species and used them to establish multispecies occupancy models. We are investigating potential solutions to resolve this problem.

(3) We used the Sørensen similarity index to measure associations between species based on support from previous literature. As counted by Mainali et al., the Sørensen index has been used in more than 700 papers across journals such as Science, Nature, and PNAS. We believe this index holds broad applicability in describing relationships between species.

(4) We agree that presenting pairwise temporal overlap in the main text would enhance clarity. We revised the manuscript to include Figure S2 in the main text and ensure that the temporal analyses are more straightforwardly presented.

Regarding the sampling collection of the scats, I'm just curious to know why you decided to use silica desiccant instead of keeping the samples frozen. I'm not familiar with this method and I guess it works fine because the environment is generally freezing cold. Yet, I would like to know more. How fresh do scat samples need to be in order to be suitable for DNA metabarcoding analyses? Additionally, what do you mean by "scats were collected within camera trapping area", could you be more specific? Have you specified a buffer around camera stations?

Thanks a lot for this specific inquiry! We refer to the scat collection method mentioned in the study of Janecka et al (2008; 2011). Silica is used to dry the scats to minimize DNA degradation. Due to the limitation of field environmental conditions, there is no suitable equipment to freeze samples during sampling, the collected scat samples should be kept dry and cool in shade, and transferred to the laboratory as soon as possible after sampling. We selected relatively fresh samples based on the color of the scat as well as broken off bits and pieces from the outside part of the scat including pieces not directly in the sun. Collect scat material about the size of a pinkie nail in the tube. If over fill the tube it will likely not dry and lead to DNA degradation.

The study area was subdivided into sample squares of 25 km2 (5×5 km) as a geographical reference for placing camera survey sites and collecting scat samples. Camera traps were set in areas believed to be important to and heavily used by wildlife, such as the bottoms of cliffs, sides of boulders, valleys and ridges along movement corridors. Also, we focused on sites with known or suspected carnivore activity to maximize probability of detection for scat samples. Therefore, transects were set around the infrared camera to collect scat samples. Length of each transect was determined by terrain, amount of scat, and available time. Each transect should have collected about 18 samples or covered 5 km of terrain to avoid uneven representation among transects and ensure that the team has sufficient time to return to base camp (Janečka et al., 2011).

Janecka J, Jackson R, Yuquang Z, Li D, Munkhtsog B, Buckley-Beason V, Murphy W. 2008. Population monitoring of snow leopards using noninvasive collection of scat samples: A pilot study. Animal Conservation 11:401–411. doi:10.1111/j.1469-1795.2008.00195.x

Janečka JE, Munkhtsog B, Jackson RM, Naranbaatar G, Mallon DP, Murphy WJ. 2011. Comparison of noninvasive genetic and camera-trapping techniques for surveying snow leopards. J Mammal 92:771–783. doi:10.1644/10-MAMM-A-036.1

Kays R, Arbogast BS, Baker‐Whatton M, Beirne C, Boone HM, Bowler M, Burneo SF, Cove MV, Ding P, Espinosa S, Gonçalves ALS, Hansen CP, Jansen PA, Kolowski JM, Knowles TW, Lima MGM, Millspaugh J, McShea WJ, Pacifici K, Parsons AW, Pease BS, Rovero F, Santos F, Schuttler SG, Sheil D, Si X, Snider M, Spironello WR. 2020. An empirical evaluation of camera trap study design: How many, how long and when? Methods Ecol Evol 11:700–713. doi:10.1111/2041-210X.13370

Regarding the discussion, the authors have information for (1) spatial distribution, (2) temporal overlap, (3) dietary requirement, they should use this information to support the discussion. Instead, sometimes it feels that authors go by exclusion or make a suggestion. For example: the authors have found dietary and temporal overlap between two apex predators (i.e., wolf and snow leopard), and they said that this suggests that spatial partitioning is responsible for their successful coexistence in this area (lines 195-196). But why "suggesting", what the co-occurrence metric says? Another example: "Apex carnivores and mesocarnivores showed substantial overlap in time overall, indicating that spatial and dietary partitioning may play a large role in facilitating their coexistence" (lines 241 - 242). However, this should not be a suggestion: your Sørensen similarity index is low proving spatial divergence. So, when data supports the hypotheses, the authors should be firmer in their discussion. Generally, when reading the discussion, it felt that a figure summarizing the partitioning would be much needed to digest which type of partitioning strategy the species are using.

Thank you for your thoughtful comments and suggestions.

(1) We appreciate your insights on the discussion section, particularly concerning the interpretation of our findings on spatial distribution, temporal and dietary overlap. We acknowledge the need for clearer interpretation of our findings. We have revised the discussion section to provide more direct support. For example, in line 294-295, we modify it as “We found dietary and temporal overlap among apex carnivores, showing that spatial partitioning is responsible for their successful coexistence in this area.” In line 341-342, we modify it as “Apex carnivores and mesocarnivores exhibited considerable overlap in time overall, showing that spatial and dietary partitioning may play a large role in facilitating their coexistence.”

(2) We appreciate your suggestion regarding the inclusion of a figure summarizing partitioning strategies among species discussed. In our study, we organized the overlap index of space, time, and diet among carnivores in Table 3, which directly reflects the overlap of carnivore species in these three dimensions by summarizing them in a single table. Additionally, Figure 3 illustrates the activity patterns and overlap among species, while Figure 4 displays the primary prey of carnivores and the frequency of food utilization.

About lines 228 - 229, just as a side note, the Pallas's cat, as the red fox, selects the environment according to a greater distribution of prey species, while also selecting primarily meadows and natural environment (Greco et al. 2022, Journal of wildlife management) additionally it is not strictly diurnal (Anile et al. 2020, Wildlife Research; Greco et al. 2022, Journal of wildlife management). Regarding the Pallas's cat and its exclusion from the temporal and spatial analyses, can you specify how many independent detection events you had?

Thanks a lot for this valuable comment!

(1) We appreciate the references to recent studies highlighting its habitat preferences and activity patterns. We have revised the manuscript to acknowledge these points and provide context regarding its habitat selection strategies. Specifically, we modify it as follow: “Pallas’s cat hunts during crepuscular and diurnal periods, inhabits meadow with greater prey abundance (Anile et al., 2021; Greco et al., 2022; Ross et al., 2019).”

(2) The low detection rate of Pallas's cat (0.072) identified by single-species occupancy model raised concerns regarding the reliability of the results. The estimated high standard errors for each environmental variable and the wide confidence intervals around the detection rate further indicated potential bias or randomness. Consequently, we made the decision to exclude the Pallas's cat data from further analysis. Upon closer examination of the Pallas's cat data, it became evident that out of 319 camera sites surveyed, only 27 sites detected the presence of Pallas's cat. Notably, only 3 out of 193 sites in Gansu Province recorded detections, while Qinghai Province had 24 detections out of 126 sites. This skewed distribution of data likely contributed to the unsatisfactory outcomes observed in our models.

About the diet and results of scat analyses, have you found any sign of intra-guild predation (i.e., apex predators that kill and sometimes consume subordinate carnivores to reduce competition), this could actually represent proof of competition and spatial overlap.

Thanks a lot for your thoughtful comments!

We observed intraguild predation in the diet of wolves and snow leopards. Specifically, we found the presence of Pallas’s cat, red fox, and Tibetan fox in the diet of wolfs, and Pallas’s cat, Eurasian Badger and Tibetan fox in the diet of snow leopard. However, these intraguild predation events accounted for only 1.89% of the diet composition of apex carnivores. We suggest that the rarity of these observations may be influenced by various factors and does not necessarily provide sufficient evidence of competition and spatial overlap. Therefore, further data collection and in-depth research are needed to better understand this phenomenon.

Some minor comments: Figure 2 is really nice, while some abbreviations are missing in the caption of Table 2.

Thank you for your feedback and positive comments on Figure 2. Unfortunately, we have removed Figure 2 from the manuscript. Due to the inclusion of prey abundance and human disturbance as occupancy covariates, these variables were derived solely from infrared camera trap data and did not encompass a comprehensive dataset across the entire national park. Therefore, we were unable to accurately spatially project for carnivore species occupancy probability in nature park.

We apologize for the oversight that the abbreviations missing in the caption of Table 2. We have added the missing abbreviations to the caption of Table 2 as follow: “Abbreviations: Disrd-distance to roads, Ele-elevation, NDVI-normalized difference vegetation index, Rix- roughness index, hdis-human disturbance.”